# HumanPlus: Humanoid Shadowing and Imitation from Humans

**Zipeng Fu**[*] **Qinqqing Zhao**[*] **Qi Wu**[*] **Gordon Wetzstein** **Chelsea Finn**

Stanford University      [*]project co-leads

https://humanoid-ai.github.io

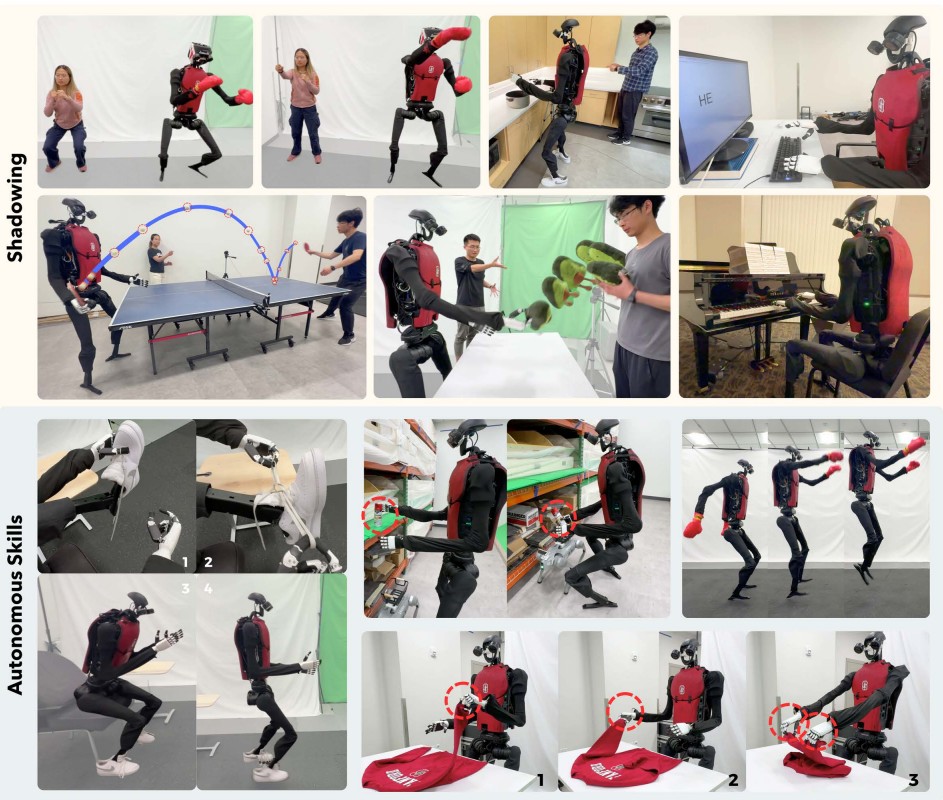

Figure 1: We present a full-stack system for humanoid robots to learn motion and autonomous skills from human data. Our system enables robots to shadow fast, diverse motions from a human operator, including boxing and playing table tennis, and to learn autonomous skills like wearing a shoe, folding clothes, and jumping high.

**Abstract:** One of the key arguments for building robots that have similar form factors to human beings is that we can leverage the massive human data for training. Yet, doing so has remained challenging in practice due to the complexities in humanoid perception and control, lingering physical gaps between humanoids and humans in morphologies and actuation, and lack of a data pipeline for humanoids to learn autonomous skills from egocentric vision. In this paper, we introduce a full-stack system for humanoids to learn motion and autonomous skills from human data. We first train a low-level policy in simulation via reinforcement learning using existing 40-hour human motion datasets. This policy transfers to the real world and allows humanoid robots to follow human body and hand motion in real time using only a RGB camera, i.e. shadowing. Through shadowing, human operators can teleoperate humanoids to collect whole-body data for learning different tasks in the real world. Using the data collected, we then perform supervised behavior cloning to train skill policies using egocentric vision, allowing humanoids to complete different tasks autonomously by imitating human skills. We demonstrate the system on our customized 33-DoF 180cm humanoid, autonomously completing tasks such as wearing a shoe to stand up and walk, unloading objects from warehouse racks, folding a sweatshirt, rearranging objects, typing, and greeting another robot with 60-100% success rates using up to 40 demonstrations.

**Keywords:** Humanoids, Whole-Body Control, Learning from Human Data

8th Conference on Robot Learning (CoRL 2024), Munich, Germany.

# 1 Introduction

Humanoid robots have long been of interest in the robotics community due to their human-like form factors. Since our surrounding environments, tasks and tools are structured and designed based on the human morphology, human-sized humanoids are the nature hardware platforms of general-purpose robots for potentially solving all tasks that people can complete. The human-like morphology of humanoids also presents a unique opportunity to leverage the vast amounts of human motion and skill data available for training, bypassing the scarcity of robot data. By mimicking humans, humanoids can potentially tap into the rich repertoire of skills and motion exhibited by humans, offering a promising avenue towards achieving general robot intelligence.

However, it has remained challenging in practice for humanoids to learn from human data. The complex dynamics and high-dimensional state and action spaces of humanoids pose difficulties in both perception and control. Traditional approaches, such as decoupling the problem into perception, planning and tracking, and separate modularization of control for arms and legs [1, 2, 3, 3], can be time-consuming to be designed and limited in scope, making them difficult to scale to the diverse range of tasks and environments that humanoids are expected to operate in. Moreover, although humanoids closely resemble humans compared to other forms of robots, physical differences between humanoids and humans in morphology and actuation still exist, including number of degrees of freedom, link length, height, weight, vision parameters and mechanisms, and actuation strength and responsiveness, presenting barriers for humanoids to effectively use and learn from human data. This problem is further exacerbated by the lack of off-the-shelf and integrated hardware platforms. Additionally, we lack an accessible data pipeline for whole-body teleoperation of humanoids, preventing researchers from leveraging imitation learning as a tool to teach humanoids arbitrary skills. Humanoids developed by multiple companies have demonstrated the potential of this data pipeline and subsequent imitation learning from the data collected, but details aren't publicly available, and autonomous demonstrations of their systems are limited to a couple tasks. Prior works use motion capture systems, first-person-view (FPV) virtual reality (VR) headsets and exoskeletons to teleoperate humanoids [4, 5, 6, 7], which are expensive and restricted in operation locations.

In this paper, we present a full-stack system for humanoids to learn motion and autonomous skills from human data. To tackle the control complexity of humanoids, we follow the recent success in legged robotics using large-scale reinforcement learning in simulation and sim-to-real transfer [8, 9] to train a low-level policy for whole-body control. Typically, learning-based low-level policies are designed to be task-specific due to time-consuming reward engineering [10, 11], enabling the humanoid hardware to demonstrate only one skill at a time, such as walking. This limitation restricts the diverse range of tasks that the humanoid platform is capable of performing. At the same time, we have a 40-hour human motion dataset, AMASS [12], that covers a wide range of skills. We leverage this dataset by first retargeting human poses to humanoid poses and then training a task-agnostic low-level policy called Humanoid Shadowing Transformer conditioning on the retargeted humanoid poses. Our pose-conditioned low-level policy transfers to the real world zero-shot.

After deploying the low-level policy that controls the humanoid given target poses, we can shadow human motion to our customized 33-DoF 180cm humanoid in real time using a single RGB camera. Using state-of-the-art human body and hand pose estimation algorithms [13, 14], we can estimate real-time human motion and retarget it to humanoid motion, which is passed as input to the low-level policy. This process is traditionally done by using motion capture systems, which are expensive and restricted in operation locations. Using line of sight, human operators standing nearby can teleoperate humanoids to collect whole-body data for various tasks in the real world, like boxing, playing the piano, playing table tennis and opening cabinets to store a heavy pot. While being teleoperated, the humanoid collects egocentric vision data through binocular RGB cameras. Shadowing provides an efficient data collection pipeline for diverse real-world tasks, bypassing the sim-to-real gap of RGB perception.

Using the data collected through shadowing, we perform supervised behavior cloning to train vision-based skill policies. A skill policy takes in humanoid binocular egocentric RGB vision as inputs and predicts the desired humanoid body and hand poses. We build upon the recent success of imitation learning from human-provided demonstrations [15, 16], and introduce a transformer-based architecture that blends action prediction and forward dynamics prediction. Using forward dynamics prediction on image features, our method shows improved performance by regularizing on image feature spaces and preventing the vision-based skill policy from ignoring image features and overfitting to proprioception. Using up to 40 demonstrations, our humanoid can autonomously complete tasks such as wearing a shoe to stand up and walk, unloading objects from warehouse

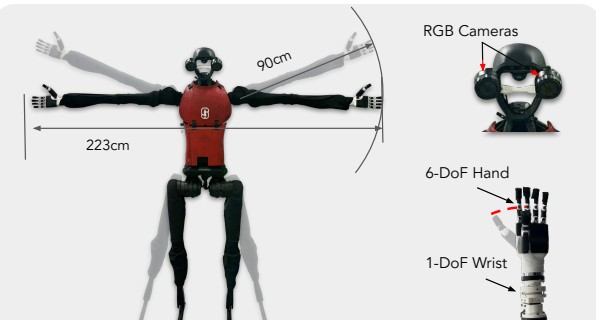

| | |
|---|---|
| Humanoid DoFs | 33 |
| Hand + Wrist DoFs | 7 (each) |
| Mass | 60kg |
| Size (cm) | 55W*30L*180H |
| Arm Payload | 7.5kg |
| Hand Payload | 1kg |
| Arm Repeatability | 5mm |
| Battery Life | ~2h (standing) |
| Max Pulling Force | 32N |

Figure 2: *Hardware Details.* Our HumanPlus robot has two egocentric RGB cameras mounted on the head, two 6-DoF dexterous hands, and 33 degrees of freedom in total.

racks, folding a sweatshirt, rearranging objects, typing, and greeting with another robot with 60-100% success rates.

The main contribution of this paper is a full-stack humanoid system for learning complex autonomous skills from human data, named HumanPlus. Core to this system is both (1) a real-time shadowing system that allows human operators to whole-body control humanoids using a single RGB camera and Humanoid Shadowing Transformer, a low-level policy that is trained on massive human motion data in simulation, and (2) Humanoid Imitation Transformer, an imitation learning algorithm that enables efficient learning from 40 demonstrations for binocular perception and high-DoF control. The synergy between our shadowing system and imitation learning algorithm allows learning of whole-body manipulation and locomotion skills directly in the real-world, such as wearing a shoe to stand up and walk, using only up to 40 demonstrations with 60-100% success.

## 2   Related Work

**Reinforcement Learning for Humanoids.**Reinforcement learning for humanoids has been predominantly focusing on locomotion. While model-based control [17, 18, 19, 20, 21, 1, 22] has made tremendous progress on a wide variety of humanoid robots [23, 24, 25, 26, 27, 28], learning-based methods can achieve robust locomotion performance for humanoids [29, 30, 31, 32, 33, 10, 34] and biped robots [35, 36, 37, 38, 39, 40, 41, 42] due to their training on highly randomized environments in simulation and their ability to adapt. Although loco-manipulation and mobile manipulation using humanoids are mostly approached via model-predictive control [43, 44, 45, 46, 47, 48], there has been some recent success on applying reinforcement learning and sim-to-real to humanoids for box relocation by explicitly modeling the scene and task in simulation [11], and for generating diverse upper-body motion [49]. In contrast, we use reinforcement leaning to train a low-level policy for task-agnostic whole-body control requiring no explicit modeling of real-world scenes and tasks in simulation.

**Teleoperation of Humanoids.**Prior works develop humanoid and dexterous teleoperation by using human motion capture suits [4, 5, 50, 51], exoskeletons [52, 53, 54, 55, 53], haptic feedback devices [56, 57, 58], and VR devices for visual feedbacks [6, 7, 59, 60] and for end-effector control [61, 62, 63, 64]. For example, Purushottam et al. develop whole-body teleoperation of a wheeled humanoid using an exoskeleton suit attached to a force plate for human motion recording. In terms of control space, prior works have done teleoperation in operation spaces [65, 5], upper-body teleoperation [66, 60], and whole-body teleoperation [67, 68, 69, 70, 59, 71, 72]. For example, He et al. use an RGB camera to capture human motion to whole-body teleoperate a humanoid. Seo et al. use VR controllers to teleoperate bimanual end-effectors and perform imitation learning on the collected data to learn static manipulation skills. In contrast, our work provides a full-stack system that consists of a low-cost whole-body teleoperation system using a single RGB camera for controlling every joint of a humanoid, enabling manipulation, squatting and walking, and an efficient imitation pipeline for learning autonomous manipulation and locomotion skills, enabling complex skills like wearing a shoe to stand up and walk.

**Robot Learning from Human Data.**Human data has been used extensively for robot learning, including for pre-training visual or intermediate representations or tasks [73, 74, 75, 76, 77, 78, 64, 79] leveraging Internet-scale data [80, 81, 82], and for imitation learning on in-domain human data [83, 84, 85, 86, 87, 88, 89, 90, 91, 92, 93, 94, 95, 96, 97]. For example, Qin et al. use in-domain human hand data for dexterous robotic hands to imitate. Recently, human data has also been used for training humanoids [49, 72]. Cheng et al. use offline human data for training humanoids to generate diverse upper-body motion, and He et al. use offline human data for training a whole-body teleoperation interface. In contrast, we use both offline human data for learning a low-level whole-

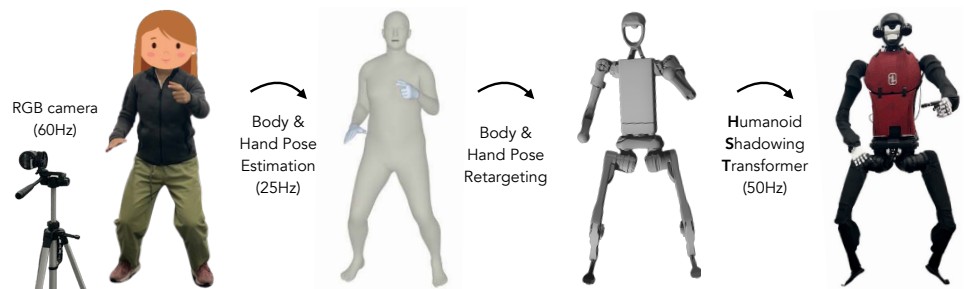

Figure 3: *Shadowing and Retargeting.* Our system uses one RGB camera for body and hand pose estimation.

body policy for real-time shadowing, and online human data through shadowing for humanoids to imitate human skills, enabling autonomous humanoid skills.

# 3 HumanPlus Hardware

Our humanoid features 33 degrees of freedom, including two 6-DoF hands, two 1-DoF wrists, and a 19-DoF body (two 4-DoF arms, two 5-DoF legs, and a 1-DoF waist), as shown on the left of Figure 2. The system is built upon the Unitree H1 robot. Each arm is integrated with an Inspire-Robots RH56DFX hand, connected via a customized wrist. Each wrist has a Dynamixel servo and two thrust bearings. Both the hands and wrists are controlled via serial communication. Our robot has two RGB webcams (Razer Kiyo Pro) mounted on its head, angled 50 degrees downward, with a pupillary distance of 160mm. The fingers can exert forces up to 10N, while the arms can hold items up to 7.5kg. The motors on legs can generate instant torques of up to 360Nm during operation. Additional technical specifications of our robot are provided on the right of Figure 2.

# 4 Human Body and Hand Data

**Offline Human Data.**We use a public optical marker-based human motion dataset, AMASS [12] to train our low-level Humanoid Shadowing Transformer. The AMASS dataset aggregates data from several human motion datasets, containing 40 hours of human motion data on a diverse ranges of tasks, and consisting of over 11,000 unique motion sequences. To ensure the quality of the motion data, we apply a filtering process based on an approach outlined in [98]. Human body and hand motions are parameterized using the SMPL-X [99] model, which includes 22 body and 30 hand 3-DoF spherical joints, 3-dimensional global translational transformation, and 3-dimensional global rotational transformation.

**Retargeting.**Our humanoid body has a subset of the degrees of freedom of SMPL-X body, consisting of only 19 revolute joints. To retarget the body poses, we copy the corresponding Euler angles from SMPL-X to our humanoid model, namely for hips, knees, ankles, torso, shoulders and elbows. Each of the humanoid hip and shoulder joints consists of 3 orthogonal revolute joints, so can be viewed as one spherical joints. Our humanoid hand has 6 degrees of freedom: 1 DoF for each of the index, middle, ring, and little fingers, and 2 DoFs for the thumb. To retarget hand poses, we map the corresponding Euler angle of each finger using the rotation of the middle joint. To compute the 1-DoF wrist angle, we use the relative rotation between the forearm and hand global orientations.

**Real-Time Body Pose Estimation and Retargeting.**To estimate human motion in the real world for shadowing, we use World-Grounded Humans with Accurate Motion (WHAM) [13] to jointly estimate the human poses and global transformations in real time using a single RGB camera. WHAM uses SMPL-X for human pose parameterization. Shown in 3, we perform real-time human-to-humanoid body retargeting using the approach described above. The body pose estimation and retargeting runs at 25 fps on an NVIDIA RTX4090 GPU.

**Real-Time Hand Pose Estimation and Retargeting.**We use HaMeR [14], a transformer-based hand pose estimator using a single RGB camera, for real-time hand pose estimation. HaMeR predicts hand poses, camera parameters, and shape parameters using the MANO [100] hand model. We perform real-time human-to-humanoid hand retargeting using the approach described above. Our hand pose estimation and retargeting runs at 10 fps on an NVIDIA RTX4090 GPU.

# 5 Shadowing of Human Motion

We formulate our low-level policy, Humanoid Shadowing Transformer, as a decoder-only transformer, shown on the left side of Figure 4. At each time step, the input to the policy is humanoid proprioception and a humanoid target pose. The humanoid proprioception contains root state (row, pitch, and base angular velocities), joint positions, joint velocities and last action. The humanoid target pose consists of target forward and lateral velocities, target roll and pitch, target yaw velocity and target joint angles, and is retargeted from a human pose sampled from the processed AMASS dataset mentioned

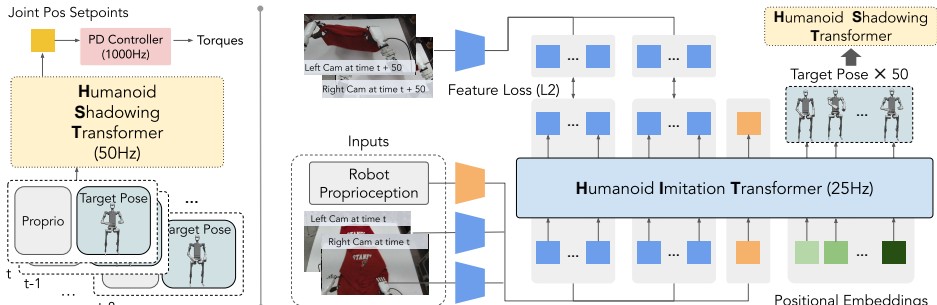

Figure 4: **Model Architectures.** Our system consists of a decoder-only transformer for low-level control, Humanoid Shadowing Transformer, and a decoder-only transformer for imitation learning, Humanoid Imitation Transformer.

in Section 4. The output of the policy is 19-dimensional joint position setpoints for humanoid body joints, which are subsequently converted to torques using a 1000Hz PD controller. The target hand joint angles are directly passed to the PD controller. Our low-level policy operates at 50Hz and has a context length of 8, so it can adapt to different environments given the observation history [31].

We use PPO [101] to train our Humanoid Shadowing Transformer in simulation by maximizing discounted expected return $\mathbb{E}\left[\sum_{t=0}^{T-1} \gamma^t r_t\right]$, where $r_t$ is the reward at time step $t$, $T$ is the maximum episode length, and $\gamma$ is the discount factor. The reward $r$ is the sum of terms encouraging matching target poses while saving energy and avoiding foot slipping. We list all the reward terms in the Table 4. We randomize the physical parameters of the simulated environment and humanoids with details in Table 5.

After training Humanoid Shadowing Transformer in simulation, we deploy it zero-shot to our humanoid in the real world for real-time shadowing. The proprioceptive observations are measured using only onboard sensors including an IMU and joint encoders. Following Section 4 and shown in Figure 3, we estimate the human body and hand poses in real time using a single RGB camera, and retarget the human poses to humanoid target poses. Illustrated in Figure 1, human operators stand near the humanoid to shadow their real-time whole-body motion to our humanoid, and use line of sight to observe the environment and behaviors of humanoid, ensuring a responsive teleoperation system. When the humanoid sits, we directly send the target poses to the PD controller, since we don't need the policy to compensate gravity, and simulating sitting with rich contacts is challenging. While being teleoperated, the humanoid collects egocentric vision data through binocular RGB cameras. Through shadowing, we provide an efficient data collection pipeline for various real-world tasks, circumventing the challenges of having realist RGB rendering, accurate soft object simulation, and diverse task specifications in simulation.

## 6 Imitation of Human Skills

Imitation learning has shown great success on learning autonomous robot skills given demonstrations on a wide range of tasks [102, 103, 15, 16, 104]. Given the real-world data collected through shadowing, we apply the same recipe to humanoids to train skill policies. We make several modifications to enable faster inference using limited onboard compute, and efficient imitation learning given binocular perception and high-DoF control.

In this work, we modify the Action Chunking Transformer [15] by removing its encoder-decoder architecture to develop a decoder-only Humanoid Imitation Transformer (HIT) for skill policies, as depicted on the right side of Figure 4. HIT processes the current image features from two egocentric RGB cameras, proprioception, and fixed positional embeddings as inputs. These image features are encoded using a pretrained ResNet encoder. Due to its decoder-only design, HIT operates by predicting a chunk of 50 target poses based on the fixed positional embeddings at the input, and it can predict tokens corresponding to the image features at their respective input positions. We incorporate an L2 feature loss on these predicted image features, compelling the transformer to predict corresponding image feature tokens for future states after execution of ground truth target pose sequences. This approach allows HIT to blend target pose prediction with forward dynamics prediction effectively. By using forward dynamics prediction on image features, our method enhances performance by regularizing image feature spaces, preventing the vision-based skill policies from ignoring image features and overfitting to proprioception. During deployment, HIT operates at 25Hz onboard, sending predicted target positions to the low-level Humanoid Shadowing Transformer asynchronously, while discarding the predicted future image feature tokens.

| | Cost | # of Operators | Whole-Body | Rearrange Objects | | | | | Rearrange Lower Objects (s) |
| | | | | Approach Object (s) | Pick Object (s) | Place Object (s) | *Whole Task (s)* | Stand (%) | |
|---|---|---|---|---|---|---|---|---|---|
| Kinesthetic | $50 | 3 | ✗ | 2.10 | 1.38 | 3.12 | 6.60 | 90.5 | - |
| ALOHA | $7050 | 2-3 | ✗ | 2.70 | 1.30 | 3.15 | 7.15 | **100** | - |
| Meta Quest | $250 | 2 | ✗ | 3.57 | 1.63 | 3.67 | 8.87 | 95.3 | - |
| Ours | $50 | 1 | ✓ | 1.76 | 0.95 | 2.59 | **5.20** | **100** | **15.34** |

Table 1: ***Teleop Comparisons & User Studies***. We report averaged completion time for 6 participants on 2 tasks.

# 7 Tasks

We select six imitation tasks and five shadowing tasks that need bimanual dexterity and whole-body control. Shown in Figure 7, these tasks span a diverse array of capabilities and objects relevant to practical applications.

In the ***Wear a Shoe and Walk*** task, the robot (1) flips a shoe, (2) picks it up, (3) puts it on, (4) presses it down to secure the fit on the left foot, (5) tangles the shoelaces with both hands, (6) grasps the right one, (7) grasps left one, (8) ties them, (9) stands up and (10) walks forward. This task demonstrates the robot's ability in complex bimanual manipulation with dexterous hands and capability in agile locomotion like standing up and walk while wearing shoes. The shoe is placed on the table uniformly randomly on a 2cm line along of robot's forward facing. Each demonstration has 1250 steps or 50 seconds.

In the ***Warehouse*** task, the robot (1) approaches the paint spray on warehouse shelves with its right hand, (2) grasps the sprayer, (3) retracts the right hand, (4) squats, (5) approaches the cart on the back of the quadruped, (6) release the spray, and (7) stands up. This tasks tests the whole-body manipulation and coordination of the robot. The standing location of the robot is randomized along a 10cm line. Each demonstration has 500 steps or 20 seconds.

In the ***Fold Clothes*** task, while maintaining balance, the robot (1) folds left sleeve, (2) folds right sleeve, (3) folds the bottom of the sweatshirt, requiring both dexterity to manipulate fabric with complex dynamics and maintaining an upright pose. The robot starts in a standing position, with a uniformaly randomly sampled root yaw deviation from +10 to -10 degrees. The sweatsheet is uniformly randomly placed with a deviation of 10cm x 10cm on the table and -30 degrees to 30 degrees in rotation. Each demonstration has 500 steps or 20 seconds.

In the ***Rearrange Objects*** task, while maintaining balance, the robot (1) approaches the object, (2) picks up the object, and (3) places the object into a basket. The complexity arises from the diverse shapes, colors, and orientations of the objects, requiring the robot to choose the appropriate hand based on the object's location and to plan its actions accordingly. In total, we uniformly sample from 4 soft objects, including stuffed toys and a ice bag, where the object is placed uniformly randomly along a 10cm line on either left or right of the basket. Each demonstration has 250 steps or 10s.

In the ***Type "AI"*** task, the robot (1) types the letter 'A', (2) releases the key, (3) types the letter 'I', and (4) releases the key. Despite being seated, the robot needs high precision in manipulation. Each demonstration has 200 steps or 8 seconds.

In the ***Two-Robot Greeting*** task, the robot (1) approaches the other robot with the correct hand after observing the other bimanual robot starts to extend one hand/arm, (2) touches the hand with the other robot, and (3) releases the hand. The other robot uniformly samples which hand to extend and stops in an end-effector region of 5cm x 5cm x 5cm. The robot needs to quickly and accurately recognize which hand to use and approaches the other robot with the correct hand while maintaining balance. Each demonstration has 125 steps or 5 seconds.

For ***Shadowing Tasks***, we demonstrate five tasks: boxing, opening a two-door cabinet to store a pot, tossing, playing the piano, playing table tennis, and typing "Hello World", showcasing the mobility and stability in shadowing fast, diverse motions and manipulating heavy objects. Videos of qualitative shadowing results can be found on the project website: https://humanoid-ai.github.io.

# 8 Experiments on Shadowing

## 8.1 Comparisons with Other Teleoperation Methods

We compare our teleoperation system with three baselines: Kinesthetic Teaching, ALOHA [15], and Meta Quest, shown in Figure 5. For ***Kinesthetic Teaching***, both arms are in passive mode and manually positioned. For ***ALOHA***, we build a pair of bimanual arms for pupputeering from two WidowX 250 robots with similar kinematic structure as our humanoid arms. For ***Meta Quest***, we use positions of the controllers for operational space control through inverse kinematics

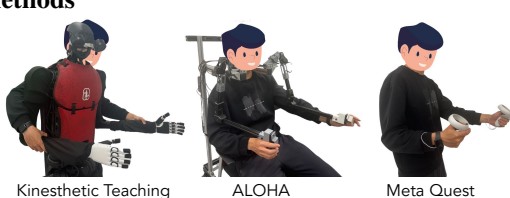

Kinesthetic Teaching     ALOHA     Meta Quest

Figure 5: ***Baseline Teleoperation Systems.***

| | Maximum Force Thresholds | | | | | Whole-Body Skills | | |
| --- | --- | --- | --- | --- | --- | --- | --- | --- |
| | forward (↑) | backward (↑) | leftward (↑) | rightward (↑) | Recovery Time (↓) | Squat(↓) | Jump(↑) | Stand Up |
| Ours | **32**N | **44**N | **70**N | **100**N | **1.2**s | **0.44**m | **0.35**m | ✓ |
| H1 Default | 24N | 36N | 40N | 40N | 15s | 0.85m | 0m | ✗ |

Table 2: ***Robustness Evaluation.*** Our low-level policy (Ours) can withstand large disturbance forces, has a shorter recovery time, and enables more whole-body skills than the manufacturer controller (H1 Default).

with gravity compensation. Shown in Table 1, all baselines do not support whole-body control and require at least two human operators for hand pose estimation. In contrast, our shadowing system simultaneously control humanoid body and hands, requiring only one human operator. Also, both *ALOHA* and *Meta Quest* are more expensive. In contrast, our system and *Kinesthetic Teaching* only require a single RGB camera.

Shown in Table 1, we conduct user studies on 6 participants to compare our shadowing system with three baselines in terms of teleoperation efficiency. Two participants have no prior teleoperation experience, while the remaining four have varying levels of expertise. None of the participants has prior experience with our shadowing system. The participants are tasked to perform the *Rearrange Objects* task and its variant, **Rearrange Lower Objects**, where an object is placed on a lower table of height 0.55m, requiring the robot to squat and thus necessitating whole-body control.

We record the average task completion time over six participants, with three trials each and three unrecorded practice rounds. We also record the average success rates of stable standing during teleoperation using our low-level policy. While *ALOHA* enables precise control of robot joint angles, its fixed hardware setup makes it harder to adapt to people with different heights and body shapes, and it does not support whole-body control of humanoids by default. *Meta Quest* often results in singularities and mismatches between target and actual poses in Cartesian space due to the limited 5 degrees of freedom of each humanoid arm plus a wrist, resulting to the longest completion time and destabilized standing at arm singularities. While *Kinesthetic Teaching* is intuitive and has a low time-to-completion, it requires multiple operators, and sometimes external forces on arms during teaching causes the humanoid to stumble. In contrast, our system has the lowest time-to-completion, has the highest success rate of stable standing, and is the only method that can be used for whole-body teleoperation, solving the *Rearrange Lower Objects* task.

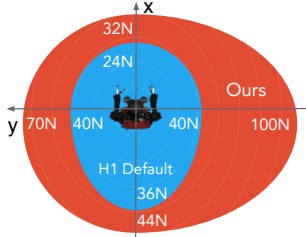

Figure 6: ***Maximum Force Thresholds.*** Our low-level policy can withstand larger forces compared to H1 Default controller.

## 8.2 Robustness Evaluation

Shown in Table 2, We evaluate our low-level policy by comparing it to the manufacturer default controller (**H1 Default**). The robot must maintain balance while manipulating with objects, so we assess robustness by applying forces to the pelvis and record the minimum forces causing instability. Shown in Figure 6, our policy withstands significantly larger forces and has a shorter recovery time. When robot is unbalanced, the manufacturer default controller takes several steps and up to 20 seconds to stabilize the robot, while ours typically recovers within one or two steps and below 3 seconds. More recovery steps result in jittery behavior and compromise manipulation performance. We also show that our policy enables more whole-body skills that are not possible by the default controller like squatting, high jumping, standing up from sitting on a chair.

## 9 Experiments on Imitation

Shown in Table 3, we compare our imitation learning method Humanoid Imitation Transformer with three baseline methods: HIT policies with monocular inputs (**Monocular**), **ACT** [15], and **Open-loop** trajectory replay, across all tasks: *Fold Clothes*, *Rearrange Objects*, *Type "AI"*, *Two-Robot Greeting*, *Warehouse*, and *Wear a Shoes and Walk*, detailed in Section 7 and Figure 7. Although each skill policy solves its task continuously autonomously without stopping, we document the success rates of consecutive sub-tasks within each task for better analysis. We conduct 10 trials per task. We calculate the success rate for a sub-task by dividing the number of successful attempts by the number of total attempts. For example in the case of *Put on Shoe* sub-task, the number of total attempts equals the number of success from the previous sub-task *Pick up Shoe*, as the robot could fail and stop at any sub-task.

Our HIT achieves higher success rates than other baselines across all tasks. Specifically, our method is only method solves the *Wear a Shoe and Walk* task, achieving success rates of 60% given 40 demonstrations, where all other methods fail. This is because our method uses binocular perception,

| | Fold Clothes (40 demos) | | | | | Rearrange Objects (30 demos) | | | | |
|---|---|---|---|---|---|---|---|---|---|---|
| | Fold Left Sleeve | Fold Right Sleeve | Fold Bottom | Stand | *Whole Task* | Approach Object | Pick up Object | Place Object | Stand | *Whole Task* |
| HIT (Ours) | 100 | 100 | 100 | 100 | **100** | 100 | 90 | 100 | 100 | **90** |
| Monocular | 80 | 50 | 100 | 100 | 40 | 80 | 88 | 100 | 100 | 70 |
| ACT | 100 | 100 | 100 | 100 | **100** | 70 | 86 | 83 | 100 | 50 |
| Open-loop | 20 | 50 | 0 | - | 0 | 0 | - | - | - | 0 |

| | Type "AI" (30 demos) | | | | | Two-Robot Greeting (30 demos) | | | | |
|---|---|---|---|---|---|---|---|---|---|---|
| | Type "A" | Leave "A" | Type "I" | Leave "I" | *Whole Task* | Approach Hand | Touch Hand | Release Hand | Stand | *Whole Task* |
| HIT (Ours) | 90 | 100 | 89 | 100 | **80** | 100 | 90 | 100 | 100 | **90** |
| Monocular | 90 | 100 | 44 | 100 | 40 | 100 | 80 | 100 | 100 | 80 |
| ACT | 30 | 20 | 0 | - | 0 | 100 | 90 | 100 | 100 | **90** |
| Open-loop | 82 | 100 | 79 | 100 | 60 | 50 | 0 | - | - | 0 |

| | Warehouse (25 demos) | | | | | | | |
|---|---|---|---|---|---|---|---|---|
| | Approach | Grasp | Retract | Squat | Approach | Release | Stand Up | *Whole Task* |
| HIT (Ours) | 100 | 90 | 100 | 100 | 100 | 100 | 100 | **90** |
| Monocular | 100 | 80 | 100 | 100 | 100 | 100 | 100 | 80 |
| ACT | 100 | 90 | 100 | 100 | 100 | 100 | 100 | **90** |
| Open-loop | 60 | 0 | - | - | - | - | - | 0 |

| | Wear a Shoe and Walk (40 demos) | | | | | | | | | | |
|---|---|---|---|---|---|---|---|---|---|---|---|
| | Flip Shoe | Pick Up Shoe | Put on Shoe | Press Shoe | Tangle Shoelaces | Grasp Right Shoelaces | Grasp Left Shoelace | Tie Shoelace | Stand Up | Walk Forward | *Whole Task* |
| HIT (Ours) | 100 | 100 | 80 | 100 | 100 | 100 | 75 | 100 | 100 | 100 | **60** |
| Monocular | 0 | - | - | - | - | - | - | - | - | - | 0 |
| ACT | 50 | 60 | 0 | - | - | - | - | - | - | - | 0 |
| Open-loop | 20 | 0 | - | - | - | - | - | - | - | - | 0 |

Table 3: ***Comparisons on Imitation.*** We show success rates of Humanoid Imitation Transformer (Ours), HIT with monocular input, ACT and open-loop trajectory replay across all tasks. Overall HIT (Ours) outperforms others.

and avoids overfitting to proprioception. *ACT* fails in *Wear a Shoe and Walk* and *Typing "AI"* tasks where it overfits to proprioception, where the robot repeatedly attempts and stucks at *Pick up Shoe* and *Leave "A"* respectively after successful completing them, avoiding uses visual feedback. *Monocular* shows lower success rates due to its lack of depth information from a single RGB camera, yielding rough interaction with the table in *Fold Clothes*. It fails the *Wear a Shoe and Walk* task completely, where depth perception is crucial. However, due to its narrower field of view, it completes some sub-tasks more successfully than other methods in the *Typing "AI"* task. *Open-loop* only works in *Typing "AI"* with no randomization, and fails in all other tasks that require reactive control.

## 10 Conclusion, Limitations and Future Directions

In this work, we present HumanPlus, a full-stack system for humanoids to learn motions and autonomous skills from human data. Throughout the development of our system, we encountered several limitations. Firstly, our hardware platform offers fewer degrees of freedom compared to human anatomy. For instance, it uses feet with 1-DoF ankles, which restricts the humanoid's ability to perform agile movements such as lifting and shaking one leg while the other remains stationary. Each arm has only 5 DoFs, including a wrist, which limits the application of 6-DoF operational space control and may result in unreachable regions during shadowing. Furthermore, the egocentric cameras are fixed on the humanoid's head and are not active, leading to a constant risk of the hands and interactions falling out of view. In addition, we currently use a fixed retargeting mapping from human poses to humanoid poses, omitting many human joints that do not exist on our humanoid hardware. This may limit the humanoids to learn from a small subset of diverse human motions. Currently, pose estimation methods do no work well given large areas of occlusion, limiting the operating regions of the human operators. Lastly, we focus on manipulation tasks with some locomotion tasks like squatting, standing up and walking in this work, as dealing with long-horizon navigation requires a much larger size of human demonstrations and accurate velocity tracking in the real world. We hope to address these limitations in future, and to enable more autonomous and robust humanoid skills that can be applied in various real-world tasks.

## Acknowledgements

We thank Steve Cousins and Oussama Khatib at Stanford Robotics Center for providing facility support for our experiments. We also thank Inspire-Robots and Unitree Robotics for providing extensive supports on hardware and low-level firmware. We thank Huy Ha, Yihuai Gao, Chong Zhang, Ziwen Zhuang, Jiaman Li, Yifeng Jiang, Yuxiang Zhang, Xingxing Wang, Tony Yang, Walter Wen, Yunguo Cui, Rosy Wang, Zhiqiang Ma, Wei Yu, Xi Chen, Mengda Xu, Peizhuo Li, Tony Z. Zhao, Lucy X. Shi and Bartie for helps on experiments, valuable discussions and supports. This project is supported by The AI Institute, ONR grant N00014-21-1-2685 and Okawa Foundation. Zipeng Fu is supported by Pierre and Christine Lamond Fellowship.

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

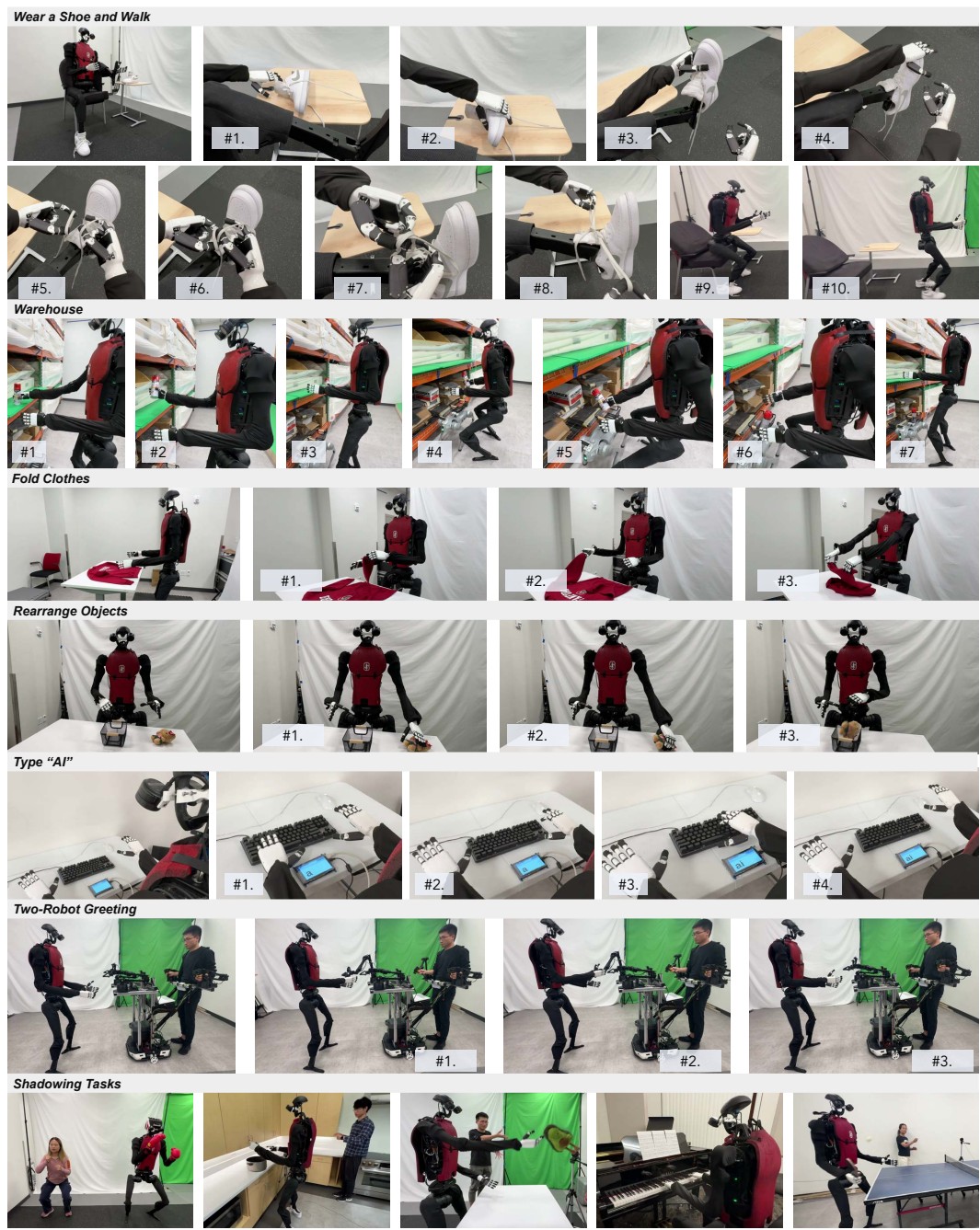

Figure 7: *Task Definitions.* We illustrate 6 autonomous tasks through imitation learning, and 5 shadowing tasks. Details are in Section 7.

| Reward Teams | Expressions |
|---|---|
| target xy velocities | $\exp(-\lvert[v_x, v_y] - [v_x^{\mathrm{tg}}, v_y^{\mathrm{tg}}]\rvert)$ |
| target yaw velocities | $\exp(-\lvert v_{\mathrm{yaw}} - v_{\mathrm{yaw}}^{\mathrm{tg}}\rvert)$ |
| target joint positions | $-\lvert q - q^{\mathrm{tg}}\rvert_2^2$ |
| target roll & pitch | $-\lvert[r, p] - [r^{\mathrm{tg}}, p^{\mathrm{tg}}]\rvert_2^2$ |
| energy | $-\lvert\tau\dot{q}\rvert_2^2$ |
| feet contact | $c == c^{\mathrm{tg}}$ |
| feet slipping | $-\lvert v_{\mathrm{feet}} \cdot \mathbb{1}[F_{\mathrm{feet}} > 1]\rvert_2$ |
| alive | $1$ |

Table 4: ***Rewards in Simulation.*** We denote $v_x$ as linear $x$ velocity, $v_y$ as linear $y$ velocity, $v_{\mathrm{yaw}}$ as angular yaw velocity, $q$ as joint positions, $\dot{q}$ as joint velocities, $r$ as roll, $p$ as pitch, $v_{\mathrm{feet}}$ as feet velocities, c as feet contact indicator, $F_{\mathrm{feet}}$ as forces on feet, and $\cdot^{\mathrm{tg}}$ as targets.

| Environment Params | Ranges |
|---|---|
| base payload | [-3.0, 3.0]kg |
| end-effector payload | [0, 0.5]kg |
| center of base mass | $[-0.1, 0.1]^3$m |
| motor strength | [0.8, 1.1] |
| friction | [0.3, 0.9] |
| control delay | [0.02, 0.04]s |

Table 5: ***Randomization in Simulation.*** We uniformly sample from these randomization ranges during training in simulation.

