# OpenReview forum: "HumanPlus: Humanoid Shadowing and Imitation from Humans"
_robot-learning.org/CoRL/2024/Conference — CoRL 2024_

### Official Review · Reviewer_uR2D · 2024-07-19
**Overall this is a solid robotics project and one of just a few approaches for humanoid mimicry and imitation learning. Many details of the approach are missing including some important information about how training is carried out.**

**Originality:** 4
**Technical Quality:** 3
**Clarity Of Presentation:** 2
**Potential Impact:** 3
**Recommendation:** 4
**Confidence:** 4

**Review:**

This is one of a few recent works to demonstrate sim2real learning of humanoid controllers for mimicking full-body human motions.

This paper goes one step further and is perhaps the first to close the loop on generating demonstrations with the controller and learning imitation policies for specific tasks. This is a natural extension of prior work to the world of humanoid robotics and represents a nice contribution.

My main criticism of the paper is that many important aspects of the approach and evaluation are not described in sufficient detail in the main paper or supplementary material. I hope that the authors will be able to include more details in the revised version. I did not find the comparison to other teleoperation methods to be very useful, so would have been happy for that to be dropped in favor of more details.

Below I'll mention some of the areas where I was wanting to know more.

Very little is said about the training process/curriculum. All I can infer is that the HST was trained on some retargeted AMASS motions. There are 11,000+ motions in that dataset and I believe that many are not appropriate/safe to attempt with a humanoid. What are the details of the training process? Is there a curriculum of motions of increasing complexity or are you actually just sampling from the 11,000 each batch of training?

Similarly little is said about the training episodes. I would assume that episodes are cutoff after falling or some other failure conditions, but no information is provided.

I couldn't find any information about adding force disturbances during training. This made it difficult to understand why the controller was shown to be more robust to disturbances compared to the manufacturer controller.

Since the HST was trained on AMAAS motions, I was expecting to see some type of evaluation on those motions. But nothing was shown or even mentioned about the AMAAS performance. Rather we are only shown shadowing results for camera driven target motions.  Overall the presented evaluation did not paint a strong picture of the capabilities of the resulting controller. A serious evaluation would have included simulation and real results on the AMASS motions --- simulation results allows for a much more robust evaluation of capabilities with the caveat that they are not fully reflective of reality.

It was surprising that the HIT controller only required the current steps information at input. This makes we wonder how robust it is to variations and whether it is primarily replaying sequences. It would have been interesting to get some information about whether including history helped or a rationale for this choice.

I was unable to get a clear picture of what the human was actually doing for shadowing finer motions involving the fingers and sitting/standing-up and walking. It would have been very useful to see the human operator during these shadowing experiments when generating data.  Is there a manual mode switch between sitting and then standing? How did the robot learn to stand --- are there instances in AMASS and if so how were they modeled? This is a very unique behavior --- going from being supported to unsupported.

The robustness experiments were not clear --- what device was used and how are forces applied?

Overall it wasn't clear how difficult it actually was to create the demonstrations for the tasks that are included. We are given a user study, but that is over a limited set of tasks.

**Quality Of The Limitations Section:**

3

**Questions For Rebuttal:**

What was the curriculum for robot learning? Was it just randomly sample a motion from the 11,000+ dataset or was there a more structured approach?

Do you have any evaluation results in simulation and real for the AMAAS training motions?

How are training episodes defined (e.g. cutoff) and are there force disturbances used during training?

What is visible in the typing task to the HIT controller? Is the robot able to see letters or is it generalizing based on relative position?

What device is used to apply forces in 8.2 and what are the details of the force application, e.g. duration and frequency?

What does the shadowing demonstration process look like for the human for the shoe task, which involves, sitting, standing, and fine manipulation? How difficult was it to generate those full demonstrations?

**Robotics Focus:**

4

**Summary Of Paper:**

The paper describes an approach for training a humanoid controller that can mimic retargeted human motions. This controller in combination with vision-based human-motion tracking is used to generate real-world ego-centric demonstrations of tasks. A transformer based architecture is then used to learning to imitate the tasks. Primarily full-body manipulation tasks are demonstrated.

**Summary Of Recommendation:**

Overall there are some impressive demonstrations here and it is perhaps the first paper to close the loop on sim2real full-body demonstration learning using a learned full-body trajectory encoder. The clarity of the paper is the biggest weakness and would even be grounds for rejection if the application novelty were not so high. The depth of evaluation could also be improved.

---

### Official Review · Reviewer_mL5C · 2024-07-21
**Impressive results, straight-forward approach, minor comments on writing**

**Originality:** 4
**Technical Quality:** 4
**Clarity Of Presentation:** 3
**Potential Impact:** 4
**Recommendation:** 4
**Confidence:** 4

**Review:**

This paper pushes the state of the art on humanoid robot control using neural networks and I recommend acceptance. The approach is straightforward and the quality of experiments and analysis in the paper is convincing. The community can benefit from the authors open-sourcing their implementation to further build on this work. The authors can also add additional details about the simulator used and training time required to train their transformer networks. I have two concerns the paper:

1. I am skeptical about the OOD extrapolation behaviors. How is the robot robust to changes in the tshirt color? Is it just replaying the motion without really understanding the visual scene? What if you change the position of the bottle in the warehouse scene, to a point that is outside the training dataset? or change the shoe? The authors should either present evidence that the policy is truly able to generalize by adding additional experiments/analysis or they should remove claims about OOD extrapolation.

2. the writing in the paper can be improved. I’ve pointed out some of the grammatical errors and typos below but I encourage the authors to proofread the paper and improve the writing.

Lines 82-90 are difficult to read and can be improved

178 - “the by”

197 - “modificationzs”

205 - “. and”

309,314,315 - “AI” opening quotes are inverted

310 - “avoids uses”

**Quality Of The Limitations Section:**

2

**Questions For Rebuttal:**

1. How long are the transformers trained for? what is the simulator used? any domain randomization?

2. What are the failure cases for the tasks shown?

3. How are the forces applied for the robustness testing?

4. What is the reason for the asymmetry in Fig. 7?

5. Thoughts on improving the jittery motion?

**Robotics Focus:**

4

**Summary Of Paper:**

This paper proposes a framework to mirror human motion without the need for expensive hardware and then uses this framework to collect demonstrations and train an autonomous control policy. The paper demonstrates the robustness and generalizability of the approach through a series impressive real robot experiments on H1 humanoid robot.

**Summary Of Recommendation:**

The paper demonstrates state-of-the-art results in challenging tasks for humanoid robots. The approach is straightforward and offers a promising alternative to expensive teleoperation hardware.

---

### Official Review · Reviewer_ycZ6 · 2024-07-22

**Originality:** 5
**Technical Quality:** 5
**Clarity Of Presentation:** 5
**Potential Impact:** 4
**Recommendation:** 4
**Confidence:** 4

**Review:**

This paper is exceptional, pulling in many recent techniques to showcase some very challenging autonomous humanoid tasks. The paper is well written and clear, the ideas presented are intuitive and brought together nicely. The diversity of tasks, including long sequences (put on a shoe) and out-of-distribution examples (rearrange objects) validates this methodology as an important work that should be explored further. The video results are very impressive.

The main criticism is that I would like to see more results that confirm design decisions (e.g. the effect of training with visual feature loss), however I understand there is already a lot in this paper and a tight page limit. I am not expecting more experiments, perhaps a follow up to this work that goes into more detail, I am curious of the next steps for the authors

For each task, e.g. wear a shoe, is it a single HIT for the entire sequence, or one for each stage of the task? How did you decide on the number of demonstrations for each?

Would it be better for operators to perform teleoperation through the sensors of the robot, rather than using line of sight to observe the behaviours? This would ensure the task is possible with restricted sensor view, e.g. hands stay in focus.

What is the effect of removing the encoder from the ACT architecture? Are there any restrictions that occur regarding diversity of demonstrations from this?

To clarify, forward dynamics prediction occurs on image features, does the HIT model reconstruct the image features at the next time step ( t+1)? What is the effect of this prediction, are you able to use fewer demonstrations and generalise better to OOD examples? Was there an ablation for this?

Would there be a benefit to combining the controllers for HIT, for example pre-train on AMASS in simulation, and fine tune on imitation? Perhaps for fine-grained tasks?

In Figure 7, it’s interesting the robot can withstand a larger force to the right (100N compared with 70N). What could be the reason for this?

What’s next, is it possible to have a single imitation transformer to scale to larger numbers of tasks? What are the limitations / benefits of this?

Minor: “put on a shoe, tie the laces, stand up and walk” is far more impressive than “wearing a shoe to stand up and walk.” Personal opinion, amazing demonstration.

“Keywords: Humnaoids” -> “Keywords: Humanoids”

“nature hardware platforms” -> “natural hardware platform”

“[1, 2, 3, 3]” -> “[1, 2, 3]”

“conditioning on the retargeted humanoid” -> “conditioned on the retargeted humanoid”

“one spherical joints.” -> “one spherical joint.”

“(row, pitch” -> “(roll, pitch”

“by maximizing the by maximizing” -> “by maximizing”

“modificationzs” -> modifications

“system simultaneously control humanoid body” -> “system simultaneously controls humanoid body

“our method is only method” -> “our method is the only method that”

Try to cite peer reviewed sources where possible, e.g:
A. Kumar, Z. Fu, D. Pathak, and J. Malik. Rma: Rapid motor adaptation for legged robots.
arXiv preprint arXiv:2107.04034, 2021. -> is from Robotics: Science and Systems

**Quality Of The Limitations Section:**

3

**Questions For Rebuttal:**

As above.

**Robotics Focus:**

4

**Summary Of Paper:**

In this paper, Human Shadowing Transformer and Human Imitation Transformer are introduced. Human Shadowing Transformer is trained in simulation to match 40 hours of human demonstrations from the AMASS dataset, including hand motions, proprioceptive only, to match reference poses. Using the shadowing transformer, task-specific data is collected (up to 40 demonstrations), and the imitation transformer is trained. This transformer is similar to action chunking with a modification to encourage image feature learning. The results on a 33 DoF humanoid, which includes 2 dexterous anthropomorphic hands, show 60-100% success rate for challenging whole-body manipulation tasks.

**Summary Of Recommendation:**

A stand out work that is beneficial for the community to expand, should be accepted.

---

### Decision · Program_Chairs · 2024-09-04

**Decision:**

Accept

**Comment:**

**Pre-rebuttal**

This paper receives three strongly positive reviews, all acknowledging the impressive and state-of-the-art demonstration of the learned humanoid controller for full-body manipulation tasks.

The main mentioned weaknesses include:
- Missing details about training and evaluation (**uR2D**, **mL5C**).
- Additional evaluation.
  - Various design decisions (**ycZ6**).
  - AMASS trained controller (**uR2D**).
- Clarification on the generalizing capability.
  - OOD extrapolation (**mL5C**).
  - Taking in only the current step information in HIT (**uR2D**).
- Unclear difficulty level of the shadowing demonstration process (**uR2D**).


---
**Post-rebuttal**

All the reviewers acknowledge the merit of the rebuttal and continue to stand by their original position of strong accept. Overall the paper has delivered solid technical contributions and impressive robot demonstration on the popular frontier of humanoid learning, which would be of wide interest to the broad CoRL audience.

Additional feedback:
- The extra details from the rebuttal on the nuance of motion shadowing of "shoelaces" and "standing up and walking" provide an important insight to the data collection process (*"To ease the human efforts during data collection ... due to hard-to-model collisions between robot hips and chairs."*). AC encourages the authors to add them into the paper.